# Synovial Fluid Interleukin-16 Contributes to Osteoclast Activation and Bone Loss through the JNK/NFATc1 Signaling Cascade in Patients with Periprosthetic Joint Infection

**DOI:** 10.3390/ijms21082904

**Published:** 2020-04-21

**Authors:** Yuhan Chang, Yi-min Hsiao, Chih-Chien Hu, Chih-Hsiang Chang, Cai-Yan Li, Steve W. N. Ueng, Mei-Feng Chen

**Affiliations:** 1Bone and Joint Research Center, Chang Gung Memorial Hospital, Taoyuan 33305, Taiwan; yhchang@cgmh.org.tw (Y.C.); r52906154@cgmh.org.tw (C.-C.H.); ccc0810.chang@gmail.com (C.-H.C.); x84301@gmail.com (C.-Y.L.); wenneng@cgmh.org.tw (S.W.N.U.); 2Department of Orthopedic Surgery, Chang Gung Memorial Hospital, Taoyuan 33305, Taiwan; 3College of Medicine, Chang Gung University, Taoyuan 33302, Taiwan; 4Graduate Institute of Clinical Medical Sciences, College of Medicine, Chang Gung University, Taoyuan 33302, Taiwan

**Keywords:** synovial fluid, interleukin-16, periprosthetic joint infection, osteoclast, bone loss, c-Jun N-terminal kinase, p38, mitogen-activated protein kinase, nuclear factor of activated T cells 1, tartrate-resistant acid phosphatase

## Abstract

Because of lipopolysaccharide (LPS)-mediated effects on osteoclast differentiation and bone loss, periprosthetic joint infection (PJI) caused by Gram-negative bacteria increases the risk of aseptic loosening after reimplantation. Synovial fluid interleukin-16 (IL-16) expression was higher in patients with PJI than in patients without joint infection. Thus, we explored the effects of IL-16 on bone. We investigated whether IL-16 modulates osteoclast or osteoblast differentiation in vitro. An LPS-induced bone loss mice model was used to explore the possible advantages of IL-16 inhibition for the prevention of bone loss. IL-16 directly activated p38 and c-Jun N-terminal kinase (JNK)/mitogen-activated protein kinase (MAPK) signaling and increased osteoclast activation markers, including tartrate-resistant acid phosphatase (TRAP), cathepsin K, and nuclear factor of activated T cells 1 (NFATc1). IL-16 directly caused monocytes to differentiate into TRAP-positive osteoclast-like cells through NFATc1 activation dependent on JNK/MAPK signaling. Moreover, IL-16 did not alter alkaline phosphatase activity or calcium deposition during osteoblastic differentiation. Finally, IL-16 inhibition prevented LPS-induced trabecular bone loss and osteoclast activation in vivo. IL-16 directly increased osteoclast activation through the JNK/NFATc1 pathway. IL-16 inhibition could represent a new strategy for treating infection-associated bone loss.

## 1. Introduction

Periprosthetic joint infection (PJI) is the most common cause of knee arthroplasty failure which accounts for 16% to 25% of all failed knee replacements [1,2,3,4] and is the third most common indication for revision hip arthroplasty [5,6,7,8,9]. Our previous findings indicated that the expression level of synovial fluid (SF) interleukin-16 (IL-16) in patients with PJI was higher than that in patients with aseptic loosening [10]. IL-16 returned to baseline values after debridement. In addition, as a lymphocyte chemoattractant factor, IL-16 is a proinflammatory chemokine for many immune cells, including cluster of differentiation 4 (CD4^+^) lymphocytes, monocytes, and eosinophils [11,12]. IL-16 is synthesized as a precursor protein (pro-IL-16) of approximately 68 kDa. Pro-IL-16 is cleaved by activated caspase-3 and then IL-16 generation [13]. CD4 serves as a signaling transducing receptor for IL-16 signaling. Increased intracytoplasmic calcium and inositol trisphosphate can be initiated when IL-16 interacts with CD4 receptors, and this subsequently activates stress-activated protein kinases (SAPKs) p46 and p54. Stimulation with IL-16 also activates members of the mitogen-activated protein kinase (MAPK) family, namely c-Jun N-terminal kinase (JNK) and p38 but does not activate extracellular signal-regulated kinase (ERK)-1 or ERK-2 [14]. IL-16 can activate monocytes and stimulate the secretion of inflammatory cytokines, chiefly tumor necrosis factor (TNF)-α, IL-1β, IL-6, and IL-15 [11,15,16]. Currently, these cytokines are believed to promote the development of septic arthritis [17,18,19]. 

PJI can result from Gram-positive (GP) or Gram-negative (GN) bacterial infections. Our previous retrospective study revealed that GN infections were associated with an increased risk of aseptic loosening as compared with GP infections [20]. The cell walls of GP bacteria contain lipoteichoic acid (LTA) components, whereas those of GN bacteria contain components of lipopolysaccharide (LPS). In studies of mice receiving intrafemoral injections of LPS or LTA, LPS reduced both the number of trabeculae and bone mineral density, whereas LTA did not have this effect [20]. In addition, disruption of the balance between bone resorption and formation has been observed in bone infections [21]. Periodic bone resorption is accomplished by osteoclasts, whereas new bone is formed by osteoblasts. Bone resorption can be regulated by differentiating new osteoclasts and activating mature osteoclasts. Osteoclasts are multinucleated cells derived from macrophages and monocytes of the hematopoietic stem cell lineage. Osteoclast differentiation and activation are mediated by both macrophage colony-stimulating factor (M-CSF) and the receptor activator of the nuclear factor-kappa B (RANK) ligand (RANKL) [22]. Osteoclasts adhere to the bone surface, where they secrete acid and proteolytic enzyme and are essential for bone resorption [23]. Osteoclasts resorb the bone matrix through acidic decalcification and proteolytic dissolution. By contrast, RANKL is typically expressed by osteoblasts and osteocytes; it binds to RANK that is present on osteoclast precursors. RANK activation promotes osteoclast survival and receptor-associated factors (TRAFs) 2, 5, and 6, which transduce signals to activate the nuclear factor kappa-light-chain-enhancer of activated B cells (NF-kB) and JNK pathways [24]. The osteoclast-specific marker genes, such as tartrate-resistant acid phosphatase (TRAP), cathepsin K, and nuclear factor of activated T cells 1 (NFATc1), are upregulated during RANKL-mediated osteoclastogenesis.

Both bacteria and bacterial infection-associated inflammation considerably influence arthroplasty outcomes. Moreover, we detected high SF IL-16 expression in patients with PJI [10]. To date, no study has examined the effects of IL-16 and bone infection on osteoblast and osteoclast activation. This study investigated whether IL-16 influences the activation of both osteoblasts and osteoclasts and whether targeting the anti-IL-16 antibody is a possible therapeutic strategy for preventing infection-associated bone loss. 

## 2. Results

### 2.1. Synovial Fluid Interleukin-16 in Patients with Periprosthetic Joint Infection Contributes to Osteoclast Activation

SF IL-16 expression in patients with joint infection was higher than that in patients with aseptic loosening (Figure 1A). After two months of antibiotic treatment and debridement, IL-16 expression levels were significantly reduced. We evaluated whether this increased IL-16 expression changed the cellular function of osteoclasts and osteoblasts. IL-16 directly induced the RAW264.7 cells to differentiate into either TRAP-positive (Figure 1B,C) or cathepsin-K-positive (Figure 1D,E) osteoclasts. Additionally, during RANKL-induced osteoclast activation, IL-16 did not alter the osteoclast-activity-associated protein or mRNA index, including TRAP and cathepsin K (Appendix A). Regarding osteoblasts, IL-16 did not change osteogenic-factor-induced MC3T3-E1 differentiation, including calcium deposition and ALP levels (Figure 1F–H). According to these findings, IL-16 directly promotes the differentiation of monocytes into osteoclasts. However, IL-16 does not regulate osteogenic-factor-induced preosteoblast differentiation or RANKL-induced osteoclast activation.

### 2.2. Effect of IL-16 on Osteoclast Activation through p38 and JNK MAPK Signaling

The RANKL-induced osteoclast activation was mediated by MAPK signaling [25,26,27,28]. Thus, we evaluated whether MAPK signaling has a role in IL-16-mediated osteoclast activation. IL-16 directly enhanced the expression of phospho-p38 and phospho-JNK MAPKs in RAW264.7 cells (Figure 2A,B). However, IL-16 did not change the expression level of phospho-ERK1/2 MAPK in RAW264.7 cells (Appendix A). Quantitative real-time PCR analysis demonstrated that IL-16 increased the transcription of p38 and JNK, as well as NFATc1 and NFATc1-regulated TRAP (Figure 2C).

### 2.3. Effect of IL-16 on TRAP-Positive Osteoclast Activation through JNK/NFATc1 Signaling Cascade

We investigated the molecular mechanism underlying the effects of JNK and p38 MAPK on the IL-16-induced increase in the number of TRAP-positive osteoclasts. Specific siRNAs for JNK and p38 MAPK were used in the investigation. The specific siRNA for JNK successfully inhibited both JNK phosphorylation and JNK mRNA expression in IL-16 stimulated RAW264.7 cells (Figure 3A,B). Moreover, siRNA-mediated JNK knockdown in RAW264.7 cell cultures attenuated subsequent NFATc1 and TRAP mRNA expression in response to IL-16 stimulation (Figure 3C). Additionally, siRNA-mediated p38 MAPK knockdown in RAW264.7 cell cultures inhibited subsequent NFATc1 but not TRAP mRNA expression in response to IL-16 stimulation (Figure 3D–F). Our data demonstrate the role of p38/JNK in IL-16 enhanced NFATc1/TRAP expression.

### 2.4. Effect of Anti-IL-16 Antibody on LPS-Induced Cathepsin K Expression and Bone Loss In Vivo

We previously demonstrated that LPSs have adverse osteoclast-mediated effects on the bone in vivo [20]. Thus, we evaluated whether anti-IL-16 antibody treatment prevents LPS-mediated cathepsin K activation and bone loss. Our histology analysis (H&E and Masson’s trichrome staining) demonstrated that the anti-IL-16 antibody significantly maintains trabecular bone density in the cross sections of femoral spongy bone (Figure 4A). LPS enhanced cathepsin K intensity, but the anti-IL-16 antibody significantly reversed this phenomenon (Figure 4). The micro-CT bone images indicated that the trabecular bone was reduced in the LPS group; however, the anti-IL-16 antibody treatment significantly maintained trabecular thickness (Tb.Th) (Figure 5). Additionally, the anti-IL-16 antibody treatment did not change the numerous LPS-reduced parameters of bone density, including the bone volume density (BS/TV), BV fraction (BV/TV), bone mineral density (BMD), and trabecular number (Tb.N). The graphical abstract is shown in Figure 6. 

## 3. Discussion

We confirmed that SF IL-16 can be used as a biomarker for PJI diagnosis [10]. To date, no studies have reported the possible effects of IL-16 on bone. Thus, studying the effect of IL-16 on both osteoclasts and osteoblasts is an interesting issue. IL-16 is synthesized by several types of cells, such as T cells, eosinophils, dendritic cells, fibroblasts, epithelial cells, and neuronal cells [11,12]. As a pleiotropic cytokine, IL-16 functions as a chemoattractant for several CD4+ immune cells. IL-16 can bind to CD4 molecules to activate monocytes and stimulate the secretion of inflammatory cytokines, namely TNF-α, IL-1β, IL-6, and IL-15 [11,15,16]. These cytokines can promote the development of septic arthritis [17,18,19]. Moreover, osteoblasts and osteoclasts play crucial roles in bone remodeling [29]. To maintain skeletal integrity, osteoclast and osteoblast activity must be precisely coordinated [30]. Osteoblasts are derived from bone marrow mesenchymal stem cells and actively promote subsequent bone formation [30]. Osteoclasts are multinucleated cells derived from macrophages and monocytes of the hematopoietic stem cell lineage [31]. Osteoclasts resorb the bone matrix through acidic decalcification and proteolytic dissolution. By contrast, RANKL is typically expressed by osteoblasts; it binds to RANK present on osteoclast precursors [22]. The RANK/RANKL signaling pathway promotes osteoclast survival and TRAFs 2, 5, and 6 that transduce signals to activate the NF-kB and JNK pathways [32]. We previously demonstrated that synovial IL-16 is a novel biomarker for PJI diagnosis [33]. Moreover, no study has analyzed the effect of IL-16 on both osteoclasts and osteoblasts. This study analyzed the effect of increased IL-16 levels on bone. We studied whether IL-16 influences osteoclast and osteoblast differentiation. This study is the first to confirm that IL-16 directly causes differentiation of monocytes into osteoclast-like cells in vitro. Additionally, IL-16 did not change the osteogenic-factor-induced osteoblast differentiation. We also confirmed that LPS does not stimulate osteoclasts or osteoblasts to produce IL-16. Although LPS did not stimulate osteoclasts or osteoblasts to increase IL-16, we confirmed that IL-16 directly caused differentiation of monocytes into osteoclast-like cells.

Osteointegration refers to the process by which bone cells have direct contact with an orthopedic implant. For successful total joint arthroplasty, long-term implant fixation is achieved through osteointegration. In addition, the current two-stage revision for PJI that involves debridement and an antibiotic-impregnated polymethylmethacrylate spacer has exhibited outcome failure [34,35]. We previously published a retrospective review of clinical records obtained from 320 patients with bacterial PJI [20]. The study demonstrated that GN bacterial infections were associated with an increased risk of aseptic loosening as compared with GP bacterial infections [20]. Moreover, LPS-treated mice had reduced body weight, higher serum osteocalcin levels, and more osteoclasts [20]. LPS accelerated differentiation of monocytes into osteoclast-like cells, whereas LTA did not [20]. Taken together, our previous findings indicate that PJI caused by GN bacteria portends a higher risk of aseptic loosening after reimplantation mainly because of the LPS-mediated effects on osteoclast differentiation [20]. Numerous studies have also confirmed that the inflammatory response caused by dead bacteria is unfavorable for osteointegration. Verdrengh et al. indicated that the postinfectious inflammatory response caused by dead bacteria prolonged bone destruction despite antibiotic treatment [36]. Rochford et al. used mouse models to assess the effects of bacterial attachment to metals on the bone healing process [37]. Their results revealed that in the group not subjected to bacteria-attached metals, fractures healed earlier than in the group subjected to bacteria-attached metals [37]. Fei et al. demonstrated that simultaneous treatment with systemic immunosuppressive agents and antibiotic therapy was beneficial for bacterial arthritis and sepsis outcomes in a mouse model [38]. Tomomatsu et al. demonstrated that LPS injection significantly reduced the BMD of the tooth socket after extraction in a murine tooth extraction model [39]. Liu et al. used LPS-doped polyethylene particles in a rat model to measure the effect of LPS on bone resorption [40]. The authors demonstrated that particle-induced impaired fixation was directly associated with increased bone resorption and repressed bone formation, supporting the clinical phenomenon of particle-related implant osteolysis and loosening [40]. Abu-Ame [41] confirmed that LPS-induced osteoclastogenesis is mediated by TNF or its p55 receptor in vivo. Because we detected increased IL-16 levels in patients with PJI [10], whether IL-16 inhibition improves GN bacterial infection-associated bone loss warrants further study. In this study, the anti-IL-16 antibody treatment prevented LPS-induced trabecular bone loss and osteoclast activation in vivo. On the basis of the aforementioned description, our findings provide a new strategy for treating PJI caused by GN bacteria. We hypothesize that treating patients with PJI with antibiotics and the anti-IL-16 antibody simultaneously reduces the incidence of aseptic loosening after reimplantation.

Nevertheless, this study also has some limitations described below. We considered animal experiments for p38/JNK mimic transfection; however, we anticipated the following potential difficulties: (1) The cost of p38 and JNK mimic drugs is high. The dosage is converted on the basis of the body weight of the mouse under study. In addition, the complete animal experiment includes an experimental group and a control group. The general laboratory cannot afford the costs of completing such an experiment. (2) The injection site for mice is a key consideration. According to this study, intraperitoneal, intravenous, intramuscular, or subcutaneous injection is not appropriate. Although intra-articular injection is the most appropriate method of injection, this technique is difficult to perform in mice. (3) Although intra-articular injections were successful, determining that the p38/JNK mimic has transfected into cells in animal models is difficult. For these reasons, we did not conduct animal experiments on p38/JNK mimic transfection. Additionally, we believe that using Western blot analysis and quantitative RT-PCR to quantify the level of NFATc1 and TRAP is valuable for this study. Because the sensitivity of quantitative RT-PCR is higher than that of Western blot analysis, first, we chose quantitative RT-PCR to verify possible mechanisms. We also used 2 × 10^5^ cells to perform Western blot analysis and quantify NFATc1/TRAP protein expression. Our experience confirmed that many cells are required to detect the NFATc1/TRAP protein through Western blot analysis. Finally, we did not examine the effect of IL-16 alone on the osteogenic ability on MC3T3-E1, because our experimental design mimicked the effects of increased IL-16 on osteoblast differentiation in periprosthetic joint infections associated with inflammation. The naive individuals did not have IL-16 (undetectable). We assumed that osteoblast precursor cells under noninflammatory condition does not encounter IL-16, and therefore we did not conduct in-depth research on this topic. 

The following points are worth considering regarding the animal model used in this study. First, we believe this animal model can simulate PJI-induced bone loss in patients because our previous retrospective study revealed that Gram-negative bacterial infections were associated with an increased risk of aseptic loosening as compared with Gram-positive bacterial infections [20]. Additionally, published data indicate that dead bacteria cause bone loss mainly through immune reactions elicited by bacterial cell wall components such as lipopolysaccharides (LPSs). Endotoxin-adherent wear particles can contribute to the aseptic loosening of orthopaedic implants even in the absence of clinical or microbiological evidence of infection. For these reasons, we used this animal model to simulate PJI-induced bone loss. Second, we initially performed intrafemoral injections of three LPS doses of 1, 10, and 20 mg/kg. Because a 1 mg/kg injection could not cause bone loss and a 20 mg/kg injection was fatal, we used 10 mg/kg LPS for the research purpose. Figure 5 indicates that the 10 mg/kg treatment reduced many bone density parameters from the seventh day to the fourteenth day, including bone volume density (BS/TV), bone volume fraction (BV/TV), bone mineral density (BMD), trabecular bone number (Tb.N), and trabecular thickness (Tb.Th). In the LPS + IL-16 antibody group, the Tb.Th value was statistically different, and also the other parameters displayed trends similar to the control group. Third, perhaps increasing the dose or frequency of IL-16 antibody injection could result in additional improvements in the other bone density parameters. Here, mice received daily intraperitoneal injections (5 μg/kg/day) of anti-IL-16 monoclonal antibody for four consecutive days. Perhaps by increasing the therapeutic dose by two times the original (10 μg/kg/day) or extending it to seven consecutive days of injection, a larger difference in bone density parameters could be observed. However, these methods of improvement must comply with the 3R (replacement, reduction, refinement) principle of animal experimentation and would also require greater labor and cost. Fourth, the area of interest quantified in this study is defined as the trabecular bone area of 1 to 3 mm^2^ which is located below the growth plate (231 slices in total). Trabecular bone is mainly distributed near the growth plate at both ends of the femoral bone; therefore, different choices of the quantitative area led to different results. Perhaps increasing the quantitative area would make it possible to observe significant differences in bone density parameters. Because we quantitatively obtained 231 slices, increasing the quantitative area would also be a considerable human resource burden. 

## 4. Materials and Methods

### 4.1. Patients and Sampling

From January 2016 to December 2017, we enrolled 33 patients (14 women and 19 men) who required hip/knee revision surgery; of the patients, 27 had PJI and 6 had aseptic loosening. PJI was diagnosed using the Musculoskeletal Infection Society (MSIS) criteria and was analyzed using the available evidence [42]. For a detailed description of PJI diagnostic methods, please refer to our previous articles [10,43]. All patients with PJI were scheduled to receive two-stage exchange arthroplasty. In brief, resection arthroplasty for PJI entailed radical debridement, prosthesis removal, antibiotic-loaded bone-cement implantation, and systemic antimicrobial agent administration to control joint infections (first-stage surgery for the infection group). The bone cements loaded with vancomycin, ceftazidime, teicoplanin, and ceftazidime engendered appropriate antibacterial effects in our hospital. Delayed prosthesis reimplantation was performed after successful antimicrobial therapy (second-stage surgery for the debridement group). During the enrollment period, six patients who had aseptic loosening and were scheduled for revision arthroplasty were enrolled as the noninfected control group. Specimens of SF were collected through needle aspiration immediately after opening the surgical wound but before arthrotomy to minimize blood contamination. Patients were followed up every 2 weeks, for 3 months, before second-stage surgery. Patients with malignant tumors or who had received immunosuppressive agents were excluded. The study was approved by the local institutional review board (IRB). Informed consent was obtained from all patients before study initiation (IRB number 105-1046C).

### 4.2. Enzyme-Linked Immunosorbent Assay

The SF specimens were delivered to the laboratory immediately after aspiration and centrifuged at 10,000× *g* to separate particulate and cellular material from each sample. The IL-16 concentrations were measured using enzyme-linked immunosorbent assay (R&D Systems, Minneapolis, MN, USA) performed in triplicate. The thresholds for intra-assay and inter-assay coefficients of variation were established at <15%, and the applicability of each candidate biomarker for PJI diagnosis was determined using receiver operating characteristic curves.

### 4.3. Osteoclast and Osteoblast Differentiation

The RAW264.7 cells were plated on a three-well immunofluorescence chamber (density of 1 × 10^4^ cells per well; Ibidi, Germany) and maintained in alpha-minimum essential medium containing 10% fetal bovine serum (FBS) and antibiotics. Cells were differentiated into mature osteoclasts in the presence of 1 ng/mL recombinant IL-16 protein (CYT-559, ProSpec, Rehovot, Israel). Mature osteoclast formation was assessed using both TRAP and cathepsin K staining. Wheat germ agglutinin was stained with the Alexa Fluor 594 conjugate (Thermo Fisher Scientific, San Jose, CA, USA), whereas 4′,6-diamidino-2-phenylindole (DAPI) staining was used for nuclei (D1306, Invitrogen, Carlsbad, CA, USA). The numbers of TRAP- or cathepsin-K-positive multinucleated (≥3 nuclei) osteoclasts were quantified under a light microscope (DFC7000 T, Leica Microsystems, Wetzlar, Germany). The steps are described as follows: First, each slide sample was taken with a 40× objective to capture three fields of view (area of 217 × 163 μm^2^) and second, the number of cells in each field was counted. Cells with a positive cathepsin K signal and nucleus number of >3 were considered osteoclasts. Other cells were classified as nonosteoclasts. Third, this experiment was repeated three times.

Calcification of MC3T3-E1 cells was assessed using Alizarin Red S staining of matrix mineralization. MC3T3-E1 cells (CRL-2593, ATCC, Manassas, VA, USA) were plated on a 12-well plate (density of 5 × 10^3^ cells per well), and then cultured with an osteogenic differentiation medium (10% FBS, 5 mM glycerol 2-phosphate, 0.1 μM dexamethasone, and 50 mM ascorbic acid). The cells were treated with or without 1 ng/mL recombinant IL-16 protein in an osteogenic differentiation medium. The medium was replaced every 3 days. After 6 weeks, the medium was removed, and the cells were washed with phosphate-buffered saline (PBS) and fixed in 10% paraformaldehyde (10 min, 25 °C). Cells were stained in a solution of 2% Alizarin Red S for 30 min at 25 °C (ScienCell, Carlsbad, CA, USA), washed three times with distilled water, and air dried before being photographed. This experiment was repeated in triplicate. Calcium assay was performed using a Calcium LiquiColor Assay (Stanbio laboratory, Boerne, TX, USA) in accordance with the manufacturer’s instructions. The MC3T3-E1 cells were plated on a 12-well plate (density of 1 × 10^4^ cells per well), and then cultured as previously described.

For alkaline phosphatase (ALP) staining, the MC3T3-E1 cells were plated on a 12-well plate (density of 1 × 10^4^ cells per well), and then cultured as previously described. The medium was removed, and the cells were washed with PBS and fixed in 60% citrate-buffered acetone. The ALP staining solution was prepared using a Fast Violet B Salt capsule (catalog no. 851-10 CAP, Sigma-Aldrich, St. Louis, MO, USA) and dissolved in naphthol AS-MX phosphate alkaline solution (catalog no. 855, Sigma-Aldrich, USA). Cells were stained in the ALP staining solution for 45 min, in the dark, at 25 °C, washed twice with distilled water, and air dried before being photographed. For the ALP assay, at 7 and 14 days, the medium was removed, and then ALP activity was measured using p-nitrophenyl phosphate as a substrate, in accordance with the method of a previous report [44]. The protein concentration was determined using Bio-Rad protein assay reagent (Bio-Rad Laboratories, Hercules, CA, USA), with bovine serum albumin as the standard.

### 4.4. Protein Extraction and Western Blot Analysis

The RAW264.7 cells were plated on a 6-well plate (density of 2 × 10^5^ cells per well) and cultured as previously described. After 24 h of incubation in a serum-free medium, cells were present in 1 ng/mL recombinant IL-16 protein. At 15, 30, 60, and 120 min, total protein extraction was performed using a protein extraction buffer (Sigma-Aldrich, USA) containing cOmplete protease inhibitor cocktail (Sigma-Aldrich, USA) and phosphatase inhibitor cocktails 2 and 3 (Sigma-Aldrich, USA). Protein samples were separated using sodium dodecyl sulfate polyacrylamide gel electrophoresis (SDS-PAGE) and transferred to a polyvinylidene fluoride (PVDF) membrane (Merck Millipore, Darmstadt, Germany). After blocking, the membrane was incubated with primary antibodies. Specific polyclonal antibodies for beta-actin (sc-47778) were purchased from Santa Cruz Biotechnology (USA); those for phospho-p38 (#9211), p38 (#8690), phospho-JNK (#4668), JNK (#9252), phospho-ERK (#4370), and ERK (#4695) MAPK were purchased from Cell Signaling Technology (USA). Finally, horseradish peroxidase conjugated secondary antibody and Western Lightning Plus-ECL reagent (Perkin-Elmer, Waltham, MA, USA) were used.

### 4.5. RNA Extraction and Quantitative Real-Time Polymerase Chain Reaction

Total RNA was extracted from cell pellets using NucleoZOL (Duren, Germany) in accordance with the manufacturer’s protocol. Real-time polymerase chain reaction (PCR) was performed using the iQ SYBR Green Supermix (Bio-Rad Laboratories, Hercules, CA, USA) for specimens and diluted standards. Serial dilutions of purified DNA were used for real-time PCR calibration. PCR-grade water was used as a negative control. The primer sequences used for amplifications were as follows: mouse glyceraldehyde 3-phosphate dehydrogenase (GAPDH), 5′-AATGGTGAAGGTCGGTGTG-3′ (forward) and 5′-GTGGAGTCATACTGGAACATGTAG-3′ (reverse); mouse TRAP, 5′-CGTCTCTGCACAGATTGCAT-3′ (forward) and 5′-GAGTTGCCACACAGCATCAC-3′ (reverse); mouse NFATc1, 5′-CTTCCAAGTTTCCACTCGGC-3′ (forward) and 5′-CGAGGTGACACTAGGGGACA-3′ (reverse). Experiments were performed in triplicate, with coefficients of variation <5%. TRAP and NFATc1 expression were normalized against GAPDH expression. 

### 4.6. siRNA Transfection

MAPK expression was silenced by small RNA interference. The p38 MAPK siRNA (#6564), JNK siRNA (#6232), and scrambled duplex control siRNA (#6568) were purchased from Cell Signaling Technology (Danvers, MA, USA) and used to transfect the cells. The siRNAs were introduced to cultured RAW264.7 cells by using the lipid-mediated transfection reagent RNAiMax (Invitrogen, Carlsbad, CA, USA) in 6-well culture dishes in accordance with the manufacturer’s instructions. The final concentration of siRNAs was 100 nmol/L.

### 4.7. Experimental Animal Studies

All animal procedures complied with the National Institutes of Health in the United States guidelines and were reviewed and approved by the local Hospital Animal Care and Use Committee (Institutional Animal Care and Use Committee approval number 2017070401, approval date: 7 September 2017). Ten-week-old male C57BL/6 mice were purchased from BioLasco Biotechnology (Taipei, Taiwan). Animals were initially anesthetized using an intraperitoneal injection of a 0.01 mL/kg mixture (1:1 *v*/*v*) of tiletamine-zolazepam (Zoletil® 50, Carros, France) and xylazine hydrochloride (Rompun® Bayer HealthCare, Berlin, Germany) and subsequently subjected to intrafemoral injections of 10 mg/kg LPS (*Escherichia coli* O127:B8, Sigma-Aldrich, St Louis, MO, USA) or vehicle PBS solution. On postinjection days 7 to 10, mice were euthanized (5 to 6 mice per group). The inoculated femur was immediately fixated in 10% formaldehyde and subjected to microcomputed tomography (micro-CT) analysis. 

### 4.8. Anti-IL-16 Antibody Treatment

After intrafemoral delivery of LPS or vehicle PBS, mice received daily intraperitoneal injections (5 μg/kg/day) of anti-IL-16 antibody (MAB1727, R&D Systems, Minneapolis, MN, USA) for 4 consecutive days.

### 4.9. Histochemistry and Immunofluorescence Staining

The femur samples were harvested in neutral buffered formalin (10%), incubated in a rapid decalcifier solution, trimmed, and embedded in paraffin. Subsequently, 4 mm thick sections were stained with (1) hematoxylin/eosin (H&E), (2) Masson’s trichrome, and (3) cathepsin K (1:100, ab19027, Cambridge, MA, Abcam). Primary antibodies against cathepsin K (1:100, ab19027, Abcam, USA) were used for immunofluorescence staining. Samples were subsequently incubated with a secondary antibody, namely Alexa Fluor 488-conjugated anti-rabbit IgG (1:500, A21206, Invitrogen, Carlsbad, CA, USA), for 60 min at 25 °C.

The signal of osteoclasts occurs around the trabecular bone area. We used MetaMorph Offline (v7.8.13.0, Molecular Devices, Sunnyvale, CA, USA) to quantify the cathepsin K signal in the trabecular bone region as follows: (1) We used the software to quantify the black area representing the trabecular bone. (2) The green fluorescent signal represented cathepsin-K-positive osteoclasts; thus, we quantified the green fluorescent signal. (3) The value obtained by dividing green fluorescence signals by the trabecular bone area represented the intensity of cathepsin-K-positive osteoclasts in the trabecular bone. 4) Each group had five or six mice, and three photos were taken in different areas in each femur for quantification.

### 4.10. Micro-CT Bone Imaging

Nondestructive ultrastructural bone analysis was performed with a SkyScan 1176 micro-CT scanner (Bruker, Kontich, Belgium). Samples were wrapped in saline-soaked gauze and subsequently scanned using a 0.5 mm aluminum filter with the following parameters: voltage, 60 kV; current, 417 μA; and exposure time, 1000 ms. Images were reconstructed to a 9 μm pixel resolution using the GUP-NRecon software (version 1.7.4.2, Bruker, Kontich, Belgium) and analyzed with the Skyscan CTAn program (version 1.15.4.0, Bruker, Kontich, Belgium). Regions of interest were defined as trabecular bone areas of 1 to 3 mm^2^ located below the growth plate (231 slices). The trabecular bone was automatically isolated with the CTAn software, which was also used to calculate both morphometric indices and the bone mineral density (BMD). The density reference was validated using BMD calibration phantoms with two different hydroxyapatite concentrations (0.25 and 0.75 g/cm^3^, respectively). Illustrative three-dimensional images were obtained with the Skyscan CTVox program (version 3.3.0, Bruker, Kontich, Belgium).

### 4.11. Statistical Analysis

All data were obtained from at least three independent experiments. Quantitative data are presented as means ± standard errors of the mean and were analyzed using two-way analyses of variance (ANOVA) followed by Bonferroni’s post hoc tests. Statistical calculations were performed using GraphPad Prism 7.04 (GraphPad Inc., San Diego, CA, USA). Two-tailed *p* values <0.05 were considered statistically significant.

## Figures and Tables

**Figure 1 ijms-21-02904-f001:**
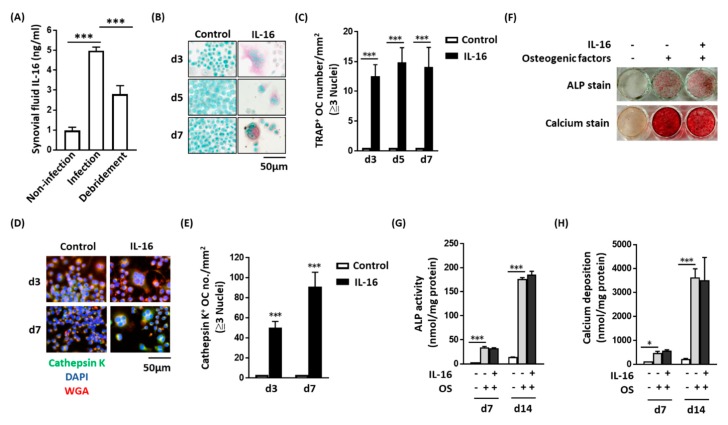
Synovial fluid interleukin-16 (IL-16) in patients with periprosthetic joint infection (PJI) contributed to osteoclast activation but osteoblast differentiation. (**A**) Synovial fluid IL-16 levels in patients with PJI (*n* = 6) were higher than those in patients with aseptic loosening (*n* = 27), the concentration of IL-16 decreased when patients received debridement surgery (*n* = 11); (**B** and **C**) IL-16 promoted RAW264.7 cell differentiation into tartrate-resistant acid phosphatase-positive osteoclast-like cells; (**D** and **E**) IL-16 promoted RAW264.7 cell differentiation into cathepsin-K-positive osteoclast-like cells; (**F**, **G**, and **H**) IL-16 did not change the expression level of ALP or calcium during osteoblast differentiation. Data are presented as means ± standard errors of the mean. Analyses were conducted with a two-way analysis of variance followed by Bonferroni’s post hoc test. ** p* < 0.05 and *** *p* < 0.001. Abbreviations: IL, interleukin; OC, osteoclast; OS, osteogenic factor; ALP, alkaline phosphatase; d, day.

**Figure 2 ijms-21-02904-f002:**
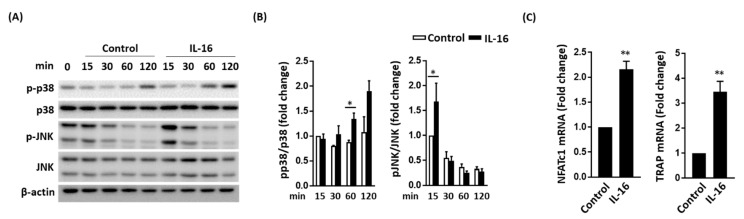
Interleukin (IL)-16 contributed to osteoclast activation through p38 and JNK MAPK signaling. (**A**,**B**) IL-16 accelerated the activation of p38 and JNK MAPK signaling; (**C**) IL-16 enhanced mRNA expression of TRAP and NFATc1. Data are presented as means ± standard errors of the mean. Analyses were conducted using a two-way analysis of variance followed by Bonferroni’s post hoc test. ** p* < 0.05 and ** *p* < 0.01. Abbreviations: IL, interleukin; pp38, phospho-p38; JNK, c-Jun N-terminal kinase; TRAP, tartrate-resistant acid phosphatase; NFATc1, nuclear factor of activated T cells 1.

**Figure 3 ijms-21-02904-f003:**
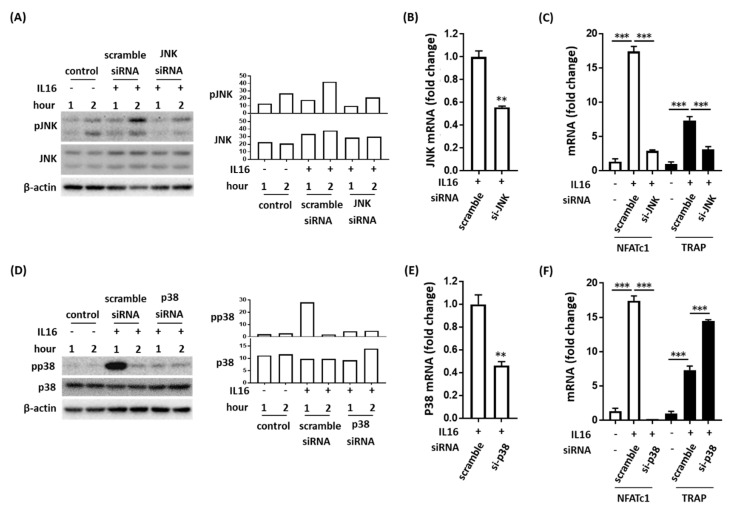
IL-16 increased the number of tartrate-resistant acid phosphatase (TRAP)-positive osteoclasts through the nuclear factor of activated T cell 1 (NFATc1) signaling pathway activated by c-Jun N-terminal kinases (JNK) but not by p38. (**A**,**B**) The specific siRNA of JNK inhibited IL-16-induced JNK phosphorylation and JNK mRNA expression in RAW264.7 cells; (**C**) The specific siRNA of JNK attenuated IL-16-induced NFATc1 and TRAP mRNA expression; (**D**,**E**) The specific siRNA of p38 MAPK inhibited IL-16-induced p38 MAPK phosphorylation and p38 MAPK mRNA expression in RAW264.7 cells; (**F**) The specific siRNA of p38 MAPK attenuated IL-16-induced NFATc1 mRNA expression and increased TRAP mRNA expression. Data are presented as means ± standard errors of the means. Analyses were conducted using two-way analysis of variance and then Bonferroni’s post hoc test. ** *p* < 0.01 and *****
*p* < 0.001.

**Figure 4 ijms-21-02904-f004:**
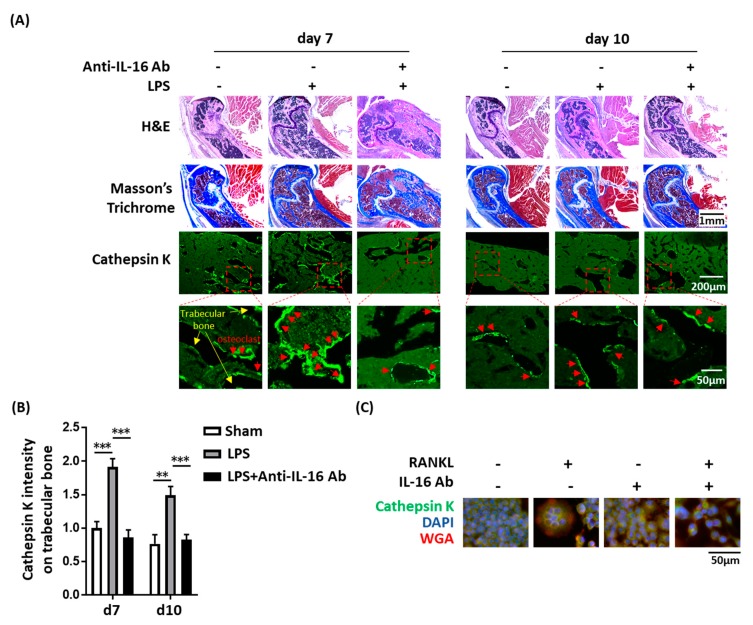
The anti-IL-16 antibody prevented LPS-induced loose trabeculae and cathepsin K expression in vivo and in vitro. (**A**) Mice received an intrafemoral injection of LPS or phosphate-buffered saline (vehicle), histological sections presented loose trabeculae in the femoral bones of LPS-treated mice, intraperitoneal injections of the anti-IL-16 antibody were performed once daily for 4 consecutive days after LPS treatment, the anti-IL-16 antibody prevented LPS-induced loose trabeculae and cathepsin K expression in vivo; (**B**) Anti-IL-16 reduced LPS-enhanced cathepsin K intensity in the femoral trabecular bone in vivo; (**C**) The IL-16 antibody inhibited RANKL-mediated RAW264.7 cell differentiation into cathepsin-K-positive osteoclast-like cells. Data are presented as means ± standard errors of the mean. Analyses were conducted using a two-way analysis of variance followed by Bonferroni’s post hoc test. **** *p* < 0.01, ***** *p* < 0.001. Abbreviations: IL, interleukin; LPS, lipopolysaccharide.

**Figure 5 ijms-21-02904-f005:**
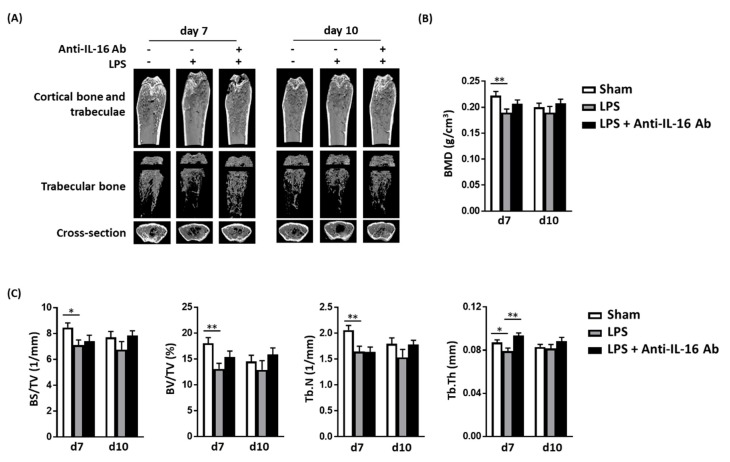
The anti-IL-16 antibody prevented LPS-induced trabecular bone loss in vivo. (**A**) Microcomputed tomography (micro-CT) revealed that the anti-IL-16 antibody reduced the number of bone trabeculae; (**B**,**C**) Quantitative results of micro-CT analysis in mice treated with or without the anti-IL-16 antibody. Anti-IL-16 antibody treatment prevented LPS-induced reductions in trabecular thickness. Data are presented as means ± standard errors of the mean. Analyses were conducted using a two-way analysis of variance followed by Bonferroni’s post hoc test. * *p* < 0.05 and **** *p* < 0.01. Abbreviations: IL, interleukin; LPS, lipopolysaccharide; Tb.Th, trabecular thickness.

**Figure 6 ijms-21-02904-f006:**
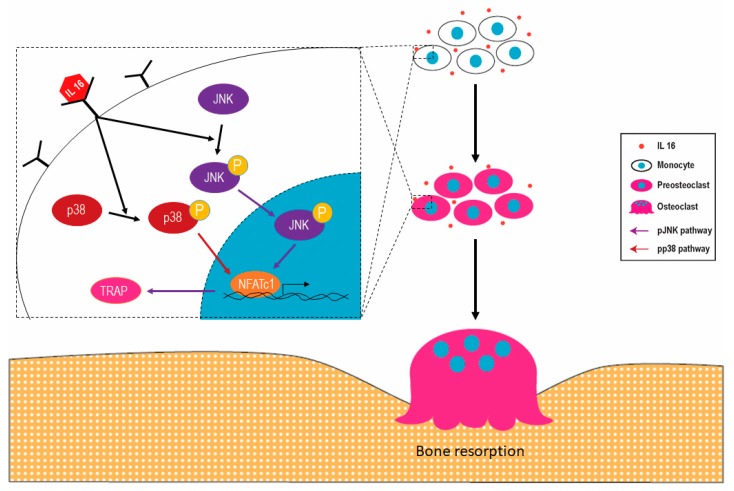
Interleukin-16 (IL-16) may promote osteoclast activation and bone loss in patients with periprosthetic joint infections through the c-Jun N-terminal kinases (JNK)/nuclear factor of activated T cells 1 (NFATc1) signaling cascade. IL-16 in inflammatory environments enhances p38 and JNK phosphorylation. The phosphorylation of JNK activates the downstream molecules NFATc1 and tartrate-resistant acid phosphatase (TRAP) and finally causes monocyte to differentiate into osteoclasts. These overactivated osteoclasts may cause bone loss in mice.

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
