# Peer review of "Synovial Fluid Interleukin-16 Contributes to Osteoclast Activation and Bone Loss through the JNK/NFATc1 Signaling Cascade in Patients with Periprosthetic Joint Infection"

_ijms, 2020, doi:10.3390/ijms21082904_

Round 1
Reviewer 1 Report
Please see the attached file.

Reviewer 2 Report
In this manuscript, the authors focused on IL-16 which has been detected in synovial fluid in periprosthetic joint infection, and examined the effect on bone cells such as osteoclasts and osteoblasts. The authors showed that using RAW264.7 cells IL-16 alone induced osteoclast differentiation while IL-16 did not show additive/enhanced osteogenic effects on MC3T3-E1. The authors showed MAPK responses by stimulation with IL-16 on RAW264.7 cells. Additionally, they addressed the effect of anti-IL-16 antibody on LPS-induced bone loss, and claimed that IL-16 inhibition prevented LPS-induced bone loss and osteoclast activation.
So far, several studies have reported molecular factors which could induce osteoclast differentiation RANKL independently (like “IL-16” in this study), although many of them were failed to be confirmed by following studies. I’d like to criticize couple of points to make sure that IL-16 alone has osteoclastogenic effects, and also that the authors’ experiments were conducted correctly.
- In this study, the authors used cell lines RAW264.7/MC3T3-E1 to examine the effect of IL-16 in vitro. They have better to test using mouse bone marrow derived monocytes/mouse calvaria or bone marrow derived osteoblasts instead of immortalized cells.
- It is surprising that IL-16 alone could induce osteoclasts. It should have to examine the effect of IL-16 on osteoclast differentiation using RANKL- and RANK-deficient cells; these experiments are required to strictly explore the effect of IL-16 on osteoclast differentiation. Additionally, the authors should show the in vitro effect of neutralization of IL-16 by anti-IL-16 antibody. This result is also required for figure 5 to explain rational of using anti-IL-16 antibody in vivo.
- The authors did not examine the effect of IL-16 alone on osteogenic ability on MC3T3-E1 (figure 1 g and h). Why don’t they examine IL-16 alone?
- In figure 2, what is the “Control”? Is it RANKL stimulation? Or just medium only? Additionally, in figure 2a and b the authors did not show 0 time control. It should be shown.
- In figure 3, the authors performed RNAi experiment using siRNAs. Message levels of JNK and p38 were inhibited (40%-60%) (figure 3b and d), but protein levels were completely same between control and siRNAs (figure 3a and d). Scramble siRNA (these should be control) induced phosphorylation of JNK and p38. These results do not make sense, and give me a doubt that the experiment did not work at all. More importantly, JNK1-/- or p38 knockdown by siRNA have already been reported to inhibit osteoclast differentiation, indicating that the experiment in figure 3 cannot explain specific importance of JNK and p38 downstream of IL-16 signaling.
- In figure 4 a, magnification is not appropriate, it is hard to see trabecular bones. It should be shown in higher magnification. Additionally, it would be better to show osteoblasts and osteoclasts numbers by H&E staining and/or TRAP staining, so that it can be determined that the effect of IL-16 on osteoclast/osteoblast differentiation in vivo.
- In figure 5, the authors mentioned that Tb.Th only showed statistical significance but other parameters (BV/TV etc.) did not show changed. This result is too weak to claim that anti-IL-16 antibody prevented LPS-induced bone loss and osteoclast activation.
Reviewer 3 Report
Chang et al investigate the effect of IL16 on both osteoclasts and osteoblast and on LPS-induced bone loss. The authors find that IL-16 is higher in PJI synovial fluids compared to synovial fluids from patients with aseptic loosening or PJI patients after debridement. The authors demonstrate that the administration of anti-IL16 antibodies ameliorate LPS-induced bone loss. However, how the anti-IL16 antibody works is not clear. More importantly, the authors propose that IL-16 induces osteoclastogenesis in a RANKL-independent manner. RANKL-independent osteoclastogenesis is receiving a lot of attention in the field of osteoclast biology. In addition, the mechanisms by which osteoclasts contribute to osseointegration in PJI are not completely determined. Thus, this manuscript covers an important area to address. However, the presented data are not sufficient to fully support the conclusion. As the authors only use the Raw267 cell line to test the effects of IL-16 on osteoclastogenesis, the authors need to modify the title to accurately reflect the conclusion obtained from their data. In general, the description of methods is not clearly written.
Major points
- The authors should test different doses of IL-16 on osteoclastogenesis.
- In Figure 2A, the authors should explain why control conditions show changes in signaling pathways.
- In Figure 2C, the authors should show the levels of NFATc1 expression by RANKL together with IL-16 to compare the efficiency of IL16 on osteoclast differentiation to RANKL-induced osteoclastogenesis. NFATc1 activation is also important for osteoclast differentiation. The authors also should show the protein expression of NFATc1.
- In Fig 3A and Fig S2A, although the authors observe the reduction of phospho-JNK or p38 and Jnk or p38 mRNA, the levels of JNK or p38 protein are comparable among all groups. The data suggest that those genes are not sufficiently knocked down. In addition, the authors should show the control conditions without IL-16 stimulation in scramble siRNA and control siRNA.
- In Fig 4, the authors conclude that treatment with the anti-IL-16 antibody prevented LPS induced bone loss. The authors should describe the method of how to calculate cathepsin K intensity in Fig.4b. To support their conclusion, the authors should perform histomorphometry analysis after TRAP staining to measure osteoclast parameters such as OC number, OC surface, and eroded surface. Please follow the ASBMR guideline.
- In Fig5, The authors should show all parameters of micro-CT including BV/TV, trabecular space, and trabecular number. Please follow the ASBMR guideline.
- The authors should show the level of serum IL-16 in a LPS-induced bone loss model.
Minor points
- The authors should describe the procedure for the debridement group in detail in the method sections. If possible, please add the list of bacteria that causes PJI in 27 PJI patients.
- In Fig 1D and E, the authors need to describe how osteoclast number was counted in the method sections.
- CD4 has been known as a receptor for IL-16. According to the previous reports (refs: Crocker, P. R et al., J Exp Med (1987) 166(2): 613-618; Anjie Zhen et al., J Virol (2014) 88(17):9934-9946)), monocytes and macrophages from humans and rats express CD4; however, CD4 is not detected on mouse peritoneal macrophages. The authors need to check the CD4 expression on the RAW264.7 cell. The authors should add their speculation about the potential mechanisms that underlie the action of IL-16 in the discussion section.
- In addition to the direct effect of IL-16 on osteoclasts, inflammatory cytokines produced by IL-16 stimulation might contribute to osteoclastogenesis. Please add this point in the discussion.
Reviewer 4 Report
The present manuscript under review titled; “Synovial Fluid Interleukin-16 Contributes to 2 Osteoclast Activation and Bone Loss Through 3 JNK/NFATc1 Signaling Cascade in Patients with 4 Periprosthetic Joint Infection” The data suggest that IL-16 directly induced osteoclast differentiation and related to the bone loss in patients with periprosthetic joint infection. In general data is appropriately interpreted and analyzed. However, there are several instances where additional data would be necessary to support the hypothesis and elevate the impact of the study.
1)Figure 1: The authors concluded that IL-16 itself have the potential to induce differentiation of RAW264.7 cells into osteoclast as well as RANKL. To strength this results, the authors need to show the effect of IL-16 on osteoclast maturation and bone resorption. Furthermore, these findings need to be confirmed in primary osteoclast cultures.
2) Figure 2: In the osteoclast formation induced by RANKL, NFATc1 expression is dependent on the TRAF6-NF-κB and c-Fos pathways. The authors need to show the effect of IL-16 on the activation of these signaling molecules.
3) Figure 3D-E: The authors showed that “siRNA-mediated p38 MAPK knockdown in RAW264.7 cell cultures inhibited subsequent NFATc1 but not TRAP mRNA expression in response to IL-16 stimulation”. Many previous studies based on the promoter analysis reported that TRAP is one of the transcriptional targets of NFATc1. Is there any other transcriptional factor which regulate TRAP expression in osteoclast formation by IL-16?
Round 2
Reviewer 2 Report
Although the authors replied/answered my questions, I’m still not convinced the biological function of IL-16 in osteoclast differentiation.
Regarding response (1 and) 7: I disagree their explanation about in vivo experiment. Even if osteoclasts mainly appear around the trabecular bone, it does not mean that BV/TV or other indices are not changed/rescued by inhibiting osteoclasts. As the authors mentioned that the experiment with anti-IL-16 antibody in LPS-induced bone loss was unsuccessful, these experimental results are not suit to demonstrate their hypothesis. The authors should address to the function of IL-16 in vivo using different experimental systems.
Regarding response 2: I do not see the revised figure in main text.
Regarding response 3: please add discussion about this issue in Discussion section.
Regarding response 4: I do not see the revised figure in main text.
Regarding response 5: I do not understand how come the authors can claim that siRNA inhibited the expression level of JNK and p38 proteins. I do not see protein reduction at all, thus I doubt that this experiment did not work.
I understood that IL-16 stimulation increased phosphorylation of JNK and p38. Thanks for the explanation. But, as protein levels of JNK and p38 were not inhibited by RNAi, I doubt that JNK and p38 is involved in downstream signaling of IL-16. Additionally, the authors did not reply my question; JNK1-/- and/or knockdown of p38 by RNAi have already been reported to inhibit osteoclast differentiation, indicating that the results in figure 3 cannot explain specific importance of JNK and p38 downstream of IL-16 signaling. What do the authors explain about this point?
Reviewer 3 Report
The authors of this manuscript have been largely responsive to the concerns from the previous review.
However, concerns remain about the data in Figure 4 and 5. The authors should move all the data in Supplementary Figure 3 to main Figure 5. Based on the results of BV/TV, anti-IL-16 antibodies did not prevent LPS-induced bone loss. The authors should amend the title of Figure 5 and the description of the data.
Given that LPS-induced Cathepsin K positive cells were suppressed by the treatment with anti-IL-16 antibodies as shown in Figure 4, it is not clear why BV/TV was not significantly restored by the treatment with anti-IL-16 antibodies and BV/TV in the treatment with anti-IL-16 antibodies was lower than non-treated conditions. The authors should discuss this point.
Reviewer 4 Report
The authors of the manuscript have adequately answered to all the raised questions. The new additions and modifications have significantly improved the quality of the manuscript.
Round 3
Reviewer 2 Report
Comments to the authors.
I have read author’s response letter very carefully. However, I do not understand nor agree the explanation for Figure 3 and Figure 5.
Regarding Figure 3, the RNAi experiment using siRNA. I still did not see any reduction of PROTEIN expression level of JNK and p38 at all. As principle of RNAi experiment is that siRNA degrades target gene mRNA, leading to reduction of PROTEIN expression, I cannot believe that RNAi experiment in Figure 3 worked.
If you would like to determine the involvement of JNK and p38 phosphorylation down-stream of IL-16 signaling, JNK kinases (such as MEK4/7) and/or p38 kinases (such as MEK3/6) would be good targets.
Regarding Figure 5, I understood that LPS-induced bone loss experiment itself worked. However, I do not agree that IL-16 antibody rescued/ameliorated LPS-induced bone loss. Indeed, the results shown in Figure 5 revealed that anti-IL-16 (with current dosage) did not affect LPS-induced bone loss. If the results could be changed by quantifying different trabecular bone area, and the results were more suit to author’s hypothesis, they should show the results. 3R (replacement, reduction, refinement) is importance to comply for all scientists. However, as long as the anti-IL-16 treatment using current dosage did not work, 3R should not be used to excuse skip experiments. If the authors would like to use current experimental results, they have to re-state all description about results of Figure 5 like that “anti-IL-16 antibody treatment with current dosage did not show ameliorative effect on LPS-induced bone loss”.